# Early-Stage Alzheimer’s Disease Categorization Using PET Neuroimaging Modality and Convolutional Neural Networks in the 2D and 3D Domains

**DOI:** 10.3390/s22124609

**Published:** 2022-06-18

**Authors:** Ahsan Bin Tufail, Nazish Anwar, Mohamed Tahar Ben Othman, Inam Ullah, Rehan Ali Khan, Yong-Kui Ma, Deepak Adhikari, Ateeq Ur Rehman, Muhammad Shafiq, Habib Hamam

**Affiliations:** 1School of Electronics and Information Engineering, Harbin Institute of Technology, Harbin 150001, China; ahsantufail@hit.edu.cn (A.B.T.); yk_ma@hit.edu.cn (Y.-K.M.); 2Department of Electrical and Computer Engineering, COMSATS University Islamabad, Sahiwal Campus, Sahiwal 57000, Pakistan; 3Registered Medical Practitioner, Pakistan Medical Commission, Islamabad 44000, Pakistan; dr.anwarnazish@gmail.com; 4Department of Computer Science, College of Computer, Qassim University, Buraydah 51452, Saudi Arabia; 5College of Internet of Things (IoT) Engineering, Changzhou Campus, Hohai University (HHU), Changzhou 213022, China; inam@hhu.edu.cn; 6Department of Electrical Engineering, University of Science and Technology Bannu, Bannu 28100, Pakistan; engr.rehan@ustb.edu.pk; 7School of Information and Software Engineering, University of Electronic Science and Technology of China, Chengdu 611731, China; deepakadhikari@std.uestc.edu.cn; 8Department of Electrical Engineering, Government College University Lahore, Lahore 54000, Pakistan; ateeq.rehman@gcu.edu.pk; 9Department of Information and Communication Engineering, Yeungnam University, Gyeongsan 38541, Korea; 10Faculty of Engineering, Université de Moncton, Moncton, NB E1A3E9, Canada; habib.hamam@umoncton.ca; 11International Institute of Technology and Management, Libreville BP1989, Gabon; 12Spectrum of Knowledge Production & Skills Development, Sfax 3027, Tunisia; 13Department of Electrical and Electronic Engineering Science, School of Electrical Engineering, University of Johannesburg, Johannesburg 2006, South Africa

**Keywords:** Alzheimer’s disease, binary classification, multiclass classification, statistical evaluation, positron emission tomography, deep learning, data augmentation

## Abstract

Alzheimer’s Disease (AD) is a health apprehension of significant proportions that is negatively impacting the ageing population globally. It is characterized by neuronal loss and the formation of structures such as neurofibrillary tangles and amyloid plaques in the early as well as later stages of the disease. Neuroimaging modalities are routinely used in clinical practice to capture brain alterations associated with AD. On the other hand, deep learning methods are routinely used to recognize patterns in underlying data distributions effectively. This work uses Convolutional Neural Network (CNN) architectures in both 2D and 3D domains to classify the initial stages of AD into AD, Mild Cognitive Impairment (MCI) and Normal Control (NC) classes using the positron emission tomography neuroimaging modality deploying data augmentation in a random zoomed in/out scheme. We used novel concepts such as the blurring before subsampling principle and distant domain transfer learning to build 2D CNN architectures. We performed three binaries, that is, AD/NC, AD/MCI, MCI/NC and one multiclass classification task AD/NC/MCI. The statistical comparison revealed that 3D-CNN architecture performed the best achieving an accuracy of 89.21% on AD/NC, 71.70% on AD/MCI, 62.25% on NC/MCI and 59.73% on AD/NC/MCI classification tasks using a five-fold cross-validation hyperparameter selection approach. Data augmentation helps in achieving superior performance on the multiclass classification task. The obtained results support the application of deep learning models towards early recognition of AD.

## 1. Introduction

Alzheimer’s Disease (AD) is a continuously evolving degenerative disturbance of the brain and a frequently occurring disorder that mostly affects the aged population worldwide [1]. It is distinguished by the continuous degeneration of nerve cells, formation of neurofibrillary tangles as well as of amyloid plaques [2]. There is a strong possibility that the cases of AD subjects may reach approximately 300 million till the year 2050 [3]. Detecting AD in its initial stages is essential to stop the degenerative process later [4]. Accurate diagnosis of AD is a cumbersome and specialized clinical task due to the ambiguity involved and the expertise required in the evaluation of the impaired responsiveness. Mild Cognitive Impairment (MCI) is a phase that occurs prior to AD and has received considerable attention in recent times.

Neuroimaging modalities, for instance Magnetic Resonance Imaging (MRI) [5] and Positron Emission Tomography (PET) [6], are playing a significant role in the recognition of AD, with several other biomarkers deployed in clinical practice [7]. PET imaging can detect Aβ deposition in the brain as well as τ-injury caused by τ plaques while neurodegeneration can be detected by the structural MRI modality.

Machine and deep learning (ML/DL) techniques have received substantial attention for the assessment of data from different sets of inputs such as text, images or volumes for different applications such as depression recognition [8], opinion leader identification [9], multi-object fuse detection [10], AD classification [11,12,13], cancer prediction [14], joint Alzheimer’s and Parkinson’s diseases classification [15,16], automatic modulation classification [17,18], diabetic retinopathy classification [19], AD assessment using independent component analysis technique [20], and endangered plant species recognition [21]. These methods can optimally infer representations from raw data through the use of a stratified sampling approach with many varying levels of intricacies. In these networks, top level topographies are more stable to the input noise than their bottom level counterparts [22,23]. Despite their strengths, problems such as overfitting, underfitting, class imbalance and missing data limit the performance of these architectures. Overfitting (due to a high dimensionality input image) is especially a well-known challenge in supervised classification that occurs when a method generates a complex separating function that cannot adequately explain a data set.

Supervised classification is important in disease diagnosis tasks such as early diagnosis of AD. Here, the aim is to search for a representation that maps the input data to a set of labels [24]. Methods such as data augmentation, architectural choices and class imbalance etc. impact the performance of an algorithm on a supervised classification task. Diagnosis of AD is naturally a multiclass problem; however, it is normally solved as genus of binary classification problems. Meaningful interpretation of features extracted by deep architectures, for instance, Convolutional Neural Networks (CNNs) for classification remains an open problem requiring further investigation and research.

Data augmentation methods such as image cropping, resizing, shifting, blurring, color jittering, and other affine transformations as well as plastic and elastic deformations etc. are routinely used in practice to enhance the size of datasets artificially which enables a DL algorithm to generalize effectively. The effect of such transformations on the applied domain enables a method to form effective representations helping it to achieve desired results optimally. In this study, we have deployed random zoomed in/out as a data augmentation method on a CNN model for multiclass classification between AD, MCI, and Normal Control (NC) classes.

Transfer learning is a meaningful way to achieve effective performances on different tasks. It has two variants, cross-domain and cross-modal. In the cross-domain variant of transfer learning, the distribution which a task belongs to is different from the original distribution of a task while in cross-modal learning it is the same. Cross-modal transfer learning has shown some remarkable successes in recent years while cross-domain transfer learning especially the one based on imagenet dataset features is a popular way of improving the performances of tasks across different applications.

Choosing optimal set of hyperparameters for DL algorithms is a difficult task due to the fact that these models usually have millions of hyperparameters. In the experiments, we chose a five-fold cross-validation (CV) approach to deal with this challenge. An advantage of this approach is that due to the small size of training and large size of validation split on a particular dataset in comparison to other CV approaches such as 10-fold CV, the obtained solution is closer to the scenarios in the real world.

The Internet of Things (IoTs) [25] has revolutionized the way things interact in health care, agriculture, signal processing, control systems, 5G and beyond 5G technologies as well as in other domains. They provided an umbrella under which sensors, software and other physical layer technologies are integrated to ease communication between devices for the provision of different services. In the medical domain, these services are critical for the early diagnosis of chronic diseases helping in disease prevention and control.

In the literature, several studies have been proposed for both AD-NC classification and MCI-to-AD prediction tasks using an iterative sparse and DL model [26], for classifying subjects into AD, MCI and NC classes using stacked denoising auto-encoders, 3D-CNNs, support vector machines (SVM), random forests, decision trees, and k-nearest neighbor classifiers [27], for AD subject classification using the gene subset from the DNA methylation dataset and enhanced deep recurrent neural network [28], for AD-NC classification using cerebral catheter angiogram neuroimages and a combination of inception version 3 and densenet201 architectures [29], as well as utilizing dysregulation patterns of miRNA biomarkers for the prediction of AD [30]. They also proposed methods for the classification of frontotemporal dementia, AD and NC using the MRI neuroimaging modality and DL models [31], a novel dense CNN network to differentiate among stable and progressive MCI classes using hippocampal morphometry [32], a U-Net styled DL architecture for AD-NC classification task from retinal vasculature images [33], an explainable 3D residual attention deep neural network (DNN) for AD-NC and progressive MCI–static MCI classification tasks [34], a multi-modal data platform architecture to implement regression tasks and to predict the progression of AD [35], as well as transfer learning models such as LeNet, AlexNet, VGG-16, VGG-19, Inception-V1, Inception-V2, Inception-V3, DenseNet-121, etc., for binary classification between NC, MCI and AD classes [36]. Similarly, research has been done to propose a deep separable CNN model along with AlexNet and GoogLeNet transfer learning based models for AD early diagnosis [37], aggregation of CNN with a deep neural network model for AD-NC classification task [38], gait-based cognitive screening and machine learning to differentiate among AD, MCI and NC classes [39], a variant of CNN for AD, NC, MCI, early MCI and late MCI classification [40], as well as an end-to-end framework comprising of CNNs and MRI scans for AD-NC binary and for multiclass tasks [41]. Furthermore, the authors have proposed a correlation analysis method for disease diagnosis between single-nucleotide polymorphisms and region of interest based on a DL model [42], transfer learning-based EfficientNet architectures for the classification of AD, MCI and NC classes [43,44], a finetuned ResNet18 network to predict MCI, early MCI, late MCI and AD classes [45], and gray level co-occurrence matrix-based features for AD-NC classification task using decision trees and support vector machine-based classifiers [46]. In addition, the authors have proposed techniques for the classification of mild dementia, moderate dementia, non-dementia and very mild dementia using deepdream, fuzzy color image enhancement, hypercolumn data enhancement techniques and support vector machine-based classifier [47], for the prediction of clinical scores of AD using group LASSO, correntropy and multilayer recurrent neural network [48], a densely connected CNN with attention mechanism for hierarchical multilevel feature extraction based on MRI images for AD-NC, MCI converter-NC and MCI converter–MCI nonconverter classification tasks [49], as well as a DL-based approach for AD-NC classification using functional MRI and PET images [50].

The up-to-date CNN architectures do not obey the Nyquist sampling theorem due to which small perturbations in input such as random shifting operation even by one pixel changes the output of the CNN architectures drastically deviating from the true target. To overcome this, blurring before subsampling is a widely deployed technique that works in such a way that blur removes above Nyquist limit frequencies, so that subsampling does not cause aliasing. To introduce this principle in modern CNN architectures, pooling operation can be performed in two steps: a stride 1 max-pooling layer followed by a stride 2 convolutional layer [51].

This study is designed to compare different DL architectures in the 2D and 3D domains. The purpose is to understand the performance differences between the architectures learned in these domains as they try to diagnose early stages of AD. To achieve this objective, we made use of CNN architectures in the 2D and 3D domains to extract features from the PET neuroimaging modality. We also deployed novel methods such as blurring before subsampling principle for AD-NC classification task and data augmentation methods such as random zoomed in/out for the multiclass classification task to study their impact on the final classification performances. This study is unique as it combines many interesting concepts in DL such as blurring before subsampling principle, cross-domain transfer learning as well as learning in the 3D domain for the early categorization of AD deploying only the PET neuroimaging modality for the first time in the literature.

The remaining contents of this paper are described next. The dataset used for this study is presented in Section 2. Methods are described in Section 3, experiments in Section 4 followed by the description of results and their discussion in Section 5. Lastly, in Section 6, we present the conclusions.

## 2. Dataset Description

We employed the PET neuroimaging modality and used scans of this modality from the AD Neuroimaging Initiative (ADNI) [52] database. Subject demographics used in this study are shown in Table 1. We presented the data in the form of mean (min-max) format. Images are used from the ADNI-1 baseline project. We used whole brain scans in our simulations to extract features that are more likely to be generalized while acknowledging that they might not be able to identify delicate changes in fine brain structures which can lead to a wrong diagnosis. Some sample scans used in the experiments are shown in Figure 1.

## 3. Methods

We used both transfer and non-transfer learning-based DL architectures, i.e., CNNs in the 2D and 3D domains to study three binary, i.e., AD/NC, AD/MCI, and NC/MCI and one multiclass, i.e., AD/NC/MCI, classification problems. We deployed custom 3D CNN architectures to study all these problems. Furthermore, We deployed a transfer learning model based on Xception architecture [53] for categorization between MCI and AD classes as well as for multiclass classification among NC, MCI and AD. A custom CNN architecture is used in the 2D domain for classification between NC and MCI classes as well as between NC and AD classes. We used blurring before subsampling principle to create the CNN architecture in the 2D domain for discrimination among NC and AD classes.

For the balanced multiclass classification task, Confusion Entropy (CEN), Relative Classifier Information (RCI), Geometric Mean (GM), Index of Balanced Accuracy (IBA), and Matthew’s Correlation Coefficient (MCC) are used as performance metrics while for the balanced binary classification tasks, sensitivity (SEN), specificity (SPEC), F_1_-score or F-measure, balanced accuracy and precision are used as performance metrics due to their versatility and ease of deployment [54].

RCI is a measure of class separability that tells us how distinctly classes have been detached from one another. Its values are in the interval between zero and one. RCI values closer to one represent better classification. CEN is a measure grounded in information theory designed to measure the performances of classifiers. CEN uses the idea of entropy and evaluates the confusion that leads to misclassified samples. Smaller values of CEN are better and highly desirable. IBA gives evidence about supremacy of a class with the optimum level of precision at an individual level. Higher values of this measure are highly desirable. GM gives information in a class coherent manner and its attention is focused only on the recall of an individual class. Higher values of GM are desirable in class discrimination tasks. MCC is the performance metric defined as the coefficient of correlation between the observed and predicted classifications. Higher values of MCC are highly desirable.

Balanced Accuracy is the recall for each class, averaged over the number of classes. It is intuitively simple as an assessment tool; the predictive quality is measured for each class independently and then aggregated. SEN or Recall represents the percentage of true positives to the true positives and false negatives summation. SPEC represents the proportion of true negatives to the summation of true negatives and false positives. Precision is the proportion of true positives to the sum of true positives and false positives. F_1_-Score or F-measure is the harmonic average of the recall and precision.

### 3.1. CNN Architectures in 2D and 3D Domains for Multiclass Categorization among AD, MCI and NC Classes

CNN architectures for multiclass classification are shown in Figure 2 and Figure 3. We chose the Xception model due to its good performance in 2D domain trained on Imagenet database. This architecture uses the idea of depthwise separable convolutions to build a robust architecture. We removed the last layers in the Xception model and embedded a flatten layer, a dense layer of 256 neurons with rectified linear unit (ReLU) activation function, a layer with a dropout probability of 0.5 and a second dense layer with 3 neurons activated by the softmax activation function to solve the multiclass classification problem.

The CNN architecture in 3D domain for multiclass classification is shown in Figure 3. An input of 79 × 95 × 69 is utilized in this architecture, which is initialized with a zero center normalization technique that divides each dimension (channel) by its standard deviation and subtracts the mean from each of these dimensions. After then, a block is repeated five times in a row. This block contains a convolutional feature extraction layer with a kernel size of 3 and 6 feature maps with a weight and bias L2 factor of 0.00005, which is used to push the weights closer to the origin. The main purpose of this convolutional layer is to extract features. Following the convolutional layer, a batch normalization (BN) layer is added to act as a regularizer and speed up the computations, followed by an exponential linear unit (ELU) activation layer with its hyperparameter α set to 1 and a maxpooling layer with filter and stride size 2 × 2 × 2, to reduce the spatial feature maps dimension and reduce the number of computations required by the network. Finally, there are three dense layers, a dropout probability of 0.1 layer, a softmax nonlinear layer, and a classification layer. To tackle the multiclass classification issue, the first fully connected (FC) layer has 100 neurons, the second has 50 neurons, and the third has three neurons. The weight and bias L2 factors of these FC layers are 0.00005.

### 3.2. CNN Architectures in 2D and 3D Domains for Binary Classification among AD and MCI Classes

CNN architectures for binary classification among AD and MCI classes are shown in Figure 4 and Figure 5.

The 2D CNN architecture for binary classification between the AD and MCI classes is shown in Figure 4. We chose the Xception model as a transfer learning model once more. We eliminated the last layers of the Xception model and replaced them with a flatten layer, an FC layer of 256 neurons, a dropout layer with probability 50%, and a second FC layer with a single neuron driven by the sigmoid function.

For this task, Figure 5 shows the CNN architecture in the 3D domain. In this architecture, an input layer accepts an input of size 79 × 95 × 69 that has been zero-centered normalized. Then, a block is repeated three times in a row. A convolutional feature extraction layer with kernel size 3 and 5 feature maps is included in this block. Following the convolutional layer, there is a non-linear activation layer with ELU non-linearity and its hyperparameter α set to 1, followed by a maxpooling dimensionality reduction layer with filter size 2 and stride size 1 to reduce the spatial dimension of feature maps, and finally a BN layer to speed up computations. Finally, there is a probability 0.5 dropout layer, followed by three FC layers, a softmax non-linear activation function layer, and a classification layer. To solve the binary classification problem, the first FC layer has 100 neurons, the second one has 10 neurons, and the third one has two neurons.

### 3.3. CNN Architectures in 2D and 3D Domains for AD-NC Binary Classification

CNN architectures for binary classification between AD and NC classes are shown in Figure 6 and Figure 7.

Figure 6 shows a 2D CNN architecture for 2-class classification between AD and NC classes. This architecture embeds the blurring before subsampling principle. The input layer in this design accepts size 79 × 95 and is initialized with the zero center normalization process. Then, a block is repeated three times in a row. The convolutional layer is trailed by a layer to implement the BN procedure, an ELU nonlinear activation layer with its hyperparameter α set to 1, a dimensionality reduction layer with maxpooling operator with filter size 2 and stride size 1 to reduce the spatial dimension of feature maps, and another convolutional feature extracting layer with kernel and stride size 2 and number of feature maps equaling 64 with a weight and bias L2 factor equal to 0.00005. Then, another block is repeated three times in a row. This block has a convolutional feature extraction layer with a kernel size of 3 and 96 feature maps with weight and bias L2 factors of 0.00005. The convolutional layer is followed by a BN layer, which is followed by a non-linear layer with an ELU activation function with its hyperparameter α set to a value of 1, which is followed by a maxpooling dimensionality reduction layer with a filter size of 2 and a stride size of 1, which is followed by another convolutional feature extracting layer with a kernel and stride size of 2 and 96 feature maps with a weight and bias L2 factor of 0.00005. Then, another block is repeated once in a row. This block has a convolutional feature extraction layer with a kernel size of 3 and 128 feature maps with weight and bias L2 factors of 0.00005. The convolutional layer is followed by a BN layer, which is followed by a non-linear layer with ELU activation function and its hyperparameter α set to 1, which is followed by a maxpooling dimensionality reduction layer with filter size 2 and stride size 1, which is followed by another convolutional feature extracting layer with kernel and stride size 2 and 128 feature maps with a weight and bias L2 factor of 0.00005. Finally, there is a global average pooling layer that averages out all features inside a feature map, followed by three dense layers, a layer with a dropout probability of 0.1, a layer with a softmax activation function, and a data classification layer. To solve the binary classification problem, the first FC layer has 300 neurons, the second has 100 neurons, and the third has 2 neurons.

Figure 7 depicts the CNN architecture for this assignment in the 3D domain. There is a layer in this design that accepts an input of 79 × 95 × 69 and is initialized with the zero center normalization process. The block is then repeated 5 times in a row. This block has four feature maps and a convolutional feature extraction layer with kernel size 3. This layer is followed by a BN layer, which functions as a regularizer and may be used to speed up calculations, a non-linear layer with an ELU activation function with its hyperparameter α set to 1, and a maxpooling dimensionality reduction layer with a filter and stride of size 2. Finally, there are three FC layers, followed by a layer with a softmax activation function and a layer that performs the final classification. There is a layer with a dropout probability of 0.1 just before the softmax layer. To tackle the binary classification problem, the first FC layer contains 100 neurons, the second contains 10 neurons, and the third contains 2 neurons.

### 3.4. CNN Architectures in 2D and 3D Domains for Binary Classification of MCI and NC

The CNN architectures for the binary classification between MCI and NC classes are given in Figure 8 and Figure 9.

Figure 8 depicts the CNN architecture in the 2D domain for this job. There is a layer in this design that accepts 79 × 95 as an input and is initialized with zero-center normalization. The block is then repeated three times in a row. This block has a convolutional feature extraction layer with a kernel size of 3 × 3 and 32 feature maps with weight and bias L2 factors of 0.00005. Succeeding the convolutional layer is a layer that normalizes the batches, which is followed by a non-linear layer with ELU activation function with its hyperparameter α set to a value of 1, which is followed by a maxpooling dimensionality reduction layer with filter and stride of size 2 to reduce the spatial dimension of feature maps. Then, another block is repeated three times in a row. This block has a convolutional feature extraction layer with a kernel size of 3 and 64 feature maps with weight and bias L2 factors of 0.00005. Succeeding the convolutional feature extracting layer is a layer that normalizes the incoming batches, followed by a non-linear layer with an ELU activation function and a hyperparameter α set to a value of 1, followed by a maxpooling dimensionality reduction layer with a filter and stride size of 2 to reduce the spatial dimension of feature maps. Then, another block is repeated three times in a row. This block has a convolutional feature extraction layer with a kernel size of 3 and 96 feature maps with weight and bias L2 factors of 0.00005. The convolutional feature extracting layer is followed by a layer that normalizes the incoming batches, followed by a layer with ELU non-linear activation function with its hyperparameter α set to 1, followed by a maxpooling dimensionality reduction layer with filter and stride size set to 2 to reduce the spatial dimension of feature maps. Then, another block is repeated once in a row. This block consists of three FC layers, a layer with a dropout probability of 0.1, a layer with a softmax activation function, and finally a classification layer. To solve the binary classification problem, the first FC layer has 100 neurons, the second has 50 neurons, and the third has 2 neurons.

Figure 9 depicts the CNN architecture in 3D for this task. There is an input layer in this design that accepts a volume of dimension 79 × 95 × 69 that has been initialized with the zero-center normalization process. Then, in a sequential method, a block is repeated 5 times. A feature extraction convolutional layer with kernel size 3 and 4 feature maps is included in this block. This layer is followed by a BN layer, a nonlinear layer with an ELU activation function with its hyperparameter α set to a value of 1, and a maxpooling dimensionality reduction layer with a filter and stride size of 2. Finally, there are three FC layers followed by a softmax activation function layer and a classification layer. There is a layer with a dropout probability of 10% immediately preceding the softmax layer. To solve the binary classification problem, the first FC layer has 100 neurons, the second has 10 neurons, and the third one has two neurons.

## 4. Experiments

To choose an optimum set of hyperparameters for the different architectures employed in this study, we employed a five-fold CV approach. An independent set is constructed for testing purposes. In the test set, there were 12 instances of NC class, 7 instances of MCI class and 4 instances of AD class. We studied balanced binary and multiclass classification problems. In the training and validation sets, we used 90 instances each of NC, MCI and AD classes.

For experiments between MCI and AD classes in the 2D domain using transfer learning, we deployed a scheduler that uses step learning rate decay as a policy for dropping the learning rate. We set the initial learning rate to 0.0001. Moreover, as an optimizer, stochastic gradient descent (SGD) is used along with categorical cross-entropy loss function. We chose a minibatch of size 16 for all the experiments. All the 2D images are preprocessed through rescaling by a factor of 1/255. The gradient clipping technique is also used that works by truncating the gradients if their magnitudes exceed a certain threshold.

For the experiments on binary categorization between NC and AD classes as well as NC and MCI classes in the 2D and 3D domains and for the MCI/AD binary classification task in the 3D domain, we deployed a scheduler that uses step learning rate decay as a policy. Adam [55] is used as an optimizer while categorical cross entropy is applied as a loss function.

For multiclass classification experiments in the 3D domain, we deployed a scheduler that uses step learning rate decay as a policy for dropping the learning rate after every epoch. We used Adam as an optimizer while categorical cross entropy is used as a loss function. We augment samples only in the training set for the experimental purpose using random zoom in/out augmentation scheme.

## 5. Results and Discussion

Result of the multiclass classification task is shown in Table 2 while those of binary classification tasks in Table 3, Table 4 and Table 5. In Table 2, Table 3, Table 4 and Table 5, it can be observed that the 3D CNN architectures outperformed their 2D counterparts in almost all performance metrics.

We deployed the blurring before subsampling principle [51] to build 2D CNN architecture for the binary classification between NC and AD classes as modern CNN architectures failed to follow the classical Nyquist sampling criterion due to which they lose the invariance of representation. However, we found that this principle has not helped the 2D domain architecture in achieving better performance in comparison to its 3D domain counterpart which could be due to weak representations learned by this architecture.

In Table 2, it can be seen that CEN (‘MCI’ class), IBA (‘MCI’ class) and GM (‘MCI’ class) for the 2D CNN architecture is better than the values of its corresponding 3D counterpart. In Table 3, Table 4 and Table 5, 3D CNN architecture has completely outperformed its 2D domain counterpart which shows the effectiveness of the representations learned by this type of architecture as these representations possess the properties of equivariance, invariance, and equivalence. An explanation for the superior performance of 3D approaches has been confirmed by [56] through their work as they found that 3D approaches outperformed their 2D counterparts comprehensively. The PET imaging modality has been shown to have better performance than MRI for AD classification task [57].

There are many reasons for the better performance of 3D CNN architectures over their 2D domain counterparts. Firstly, 3D CNN models incorporate spatial dimension during training which allows them to form representations during training. On the other hand, the 2D CNN architectures does not use this information during training but only in inference mode. Secondly, the architectural differences and hyperparameters (such as learning rate, batch size, filter numbers etc.) differed in both these architectures, which has an impact on final performances. The blurring before subsampling principle also has an impact on the performance of 2D CNN architecture for AD-NC binary classification task as it has lowered the performance of this architecture in comparison to its 3D domain counterpart which could be due to strided convolution in the architecture as convolution with strides leave information which are exploited by the 3D CNN architectures.

An evaluation free of any bias is essential in any DL classification task. This is especially important for the experiments in the medical domain because the DL architectures offer extreme flexibility in the choice of hyperparameters and other architectural choices. A major restraint of this study has been the absence of testing on cross-modal datasets. Another limitation is the choice of hyperparameters for the experiments as we cannot eliminate the possibility that other architectural choices might have achieved better performances than our own. This problem can be resolved by continuous testing of hyperparameters which is a computationally demanding task.

Transfer learning architectures that are employed in this study have inferior performance to the architectures that are trained from scratch, for instance, 3D CNN architectures. We used cross-domain transfer learning utilizing the features of Imagenet dataset which are generalized across many different domains and datasets such as natural images. One reason for the superior performance of architectures learned from scratch is that they learn input distributions better when compared with transfer learning methods. Since we only fine-tuned our transfer learning architectures, therefore, the features that are already fixed in the earlier layers due to Imagenet based training were not very helpful in obtaining better classification performances.

We found dataset augmentation to be very helpful in obtaining better classification performances on the multiclass classification problem. Augmentation synthetically increases the size of the dataset helping DL models to perform better as these systems are data-hungry and generalize better when they are presented with a large number of samples. However, data augmentation may not always help as we learned in this study that for the binary classification tasks, it actually results in the inferior performances of architectures. Caution should be exercised in the use of data augmentation methods so that maximum benefits may be reaped due to their usage.

We deployed pre-processed rather than raw data due to its superior performance as shown in [56,58,59]. In Table 2, Table 3, Table 4 and Table 5, we can see that the discrimination between NC and AD classes is easier in comparison to the one between NC and MCI classes or MCI and AD classes. One reason for this behavior is that MCI lies on the continuum between NC and AD classes and it is relatively difficult to discriminate it from NC or AD classes. As a matter of fact, MCI class instances can be further divided into two subcategories, namely progressive MCI and static MCI, where progressive MCI represent cases that progress to AD after a certain period of time while static MCI are cases that do not do so.

Table 6 presents an assessment of our work with other approaches described in the literature. Bold values in Table 6 represent our approach. As can be in Table 6, our approach using 3D CNN architectures outperformed other approaches on both binary classification between AD/NC, AD/MCI and AD/NC/MCI classification tasks. However, for NC/MCI binary classification task, the obtained results are not better than the corresponding approaches. For NC/MCI binary classification task, we observed that specialized features and the use of more data samples has brought dividends in terms of better performances on this task.

Another point worth noting is that the NC/MCI binary classification task may benefit strongly from learning from hand-crafted features than more generalized ones. One reason for this behavior could be that MCI class instances are closer in trajectory to NC class instances than to AD class instances, helping them to benefit from a design that offers the flexibility of allowing stronger correlation with the NC class instances. This point warrants further investigation.

## 6. Conclusions

In this paper, we compared and contrasted the performance of CNN architectures in 2D and 3D domains for the early detection of AD. We performed three binaries: AD/NC, NC/MCI, and AD/MCI and one multiclass AD/NC/MCI classification task using the PET neuroimaging modality and whole brain images by deploying a five-fold CV approach for hyperparameter selection. We used both transfer learning approaches (Xception architecture) as well as approaches that used training from scratch. We achieved an accuracy of 89.21% on AD/NC, 71.70% on AD/MCI, 62.25% on NC/MCI and 59.73% on AD/NC/MCI classification tasks using 3D-CNN architectures. We found the performance of CNN architectures in the 3D domain to be the best, which shows the importance of learning in the higher domain while training from scratch. We also found that the random zoomed in/out data augmentation method helps in achieving better performance for the multiclass classification task which indicates the importance of data augmentation for this problem. We also confirm that the blurring before subsampling principle does not help us in getting good classification performance. Challenges still exist in getting optimal performances such as a limited number of samples and generalization to unseen examples, etc., which can hamper the performance of classification architectures towards the early diagnosis of AD. We intend to expand on this research in the future by employing novel architectures and data augmentation approaches such as graph convolutional networks and generative adversarial networks, as well as learning in other domains such as the frequency domain using the discrete cosine transform technique.

## Figures and Tables

**Figure 1 sensors-22-04609-f001:**
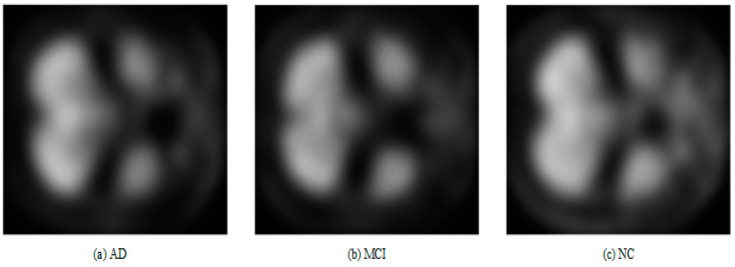
Sample PET scans used in the experiments.

**Figure 2 sensors-22-04609-f002:**
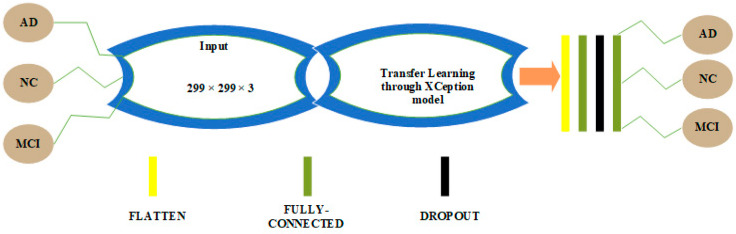
Architecture for multiclass classification in the 2D domain.

**Figure 3 sensors-22-04609-f003:**
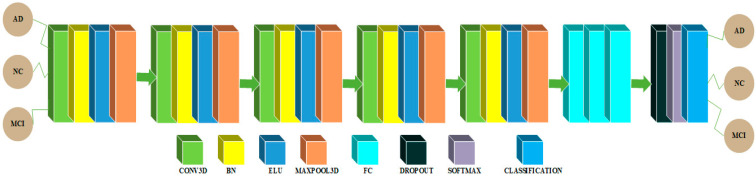
Architecture for multiclass classification in the 3D domain.

**Figure 4 sensors-22-04609-f004:**
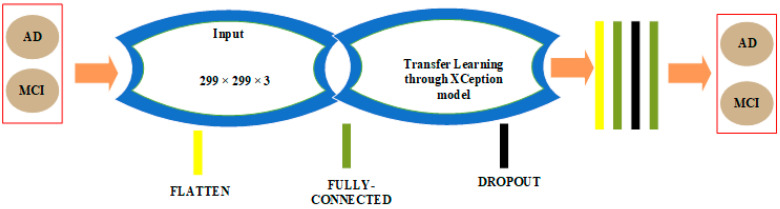
AD/MCI binary classification architecture in the 2D domain.

**Figure 5 sensors-22-04609-f005:**
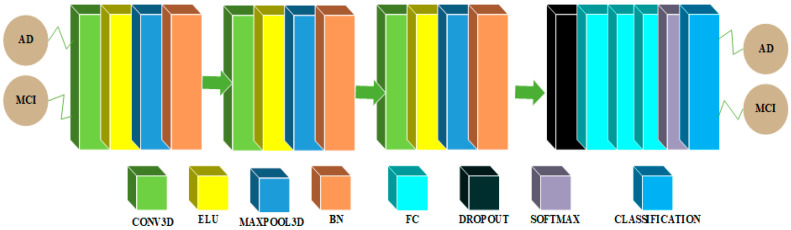
AD/MCI binary classification architecture in the 3D domain.

**Figure 6 sensors-22-04609-f006:**
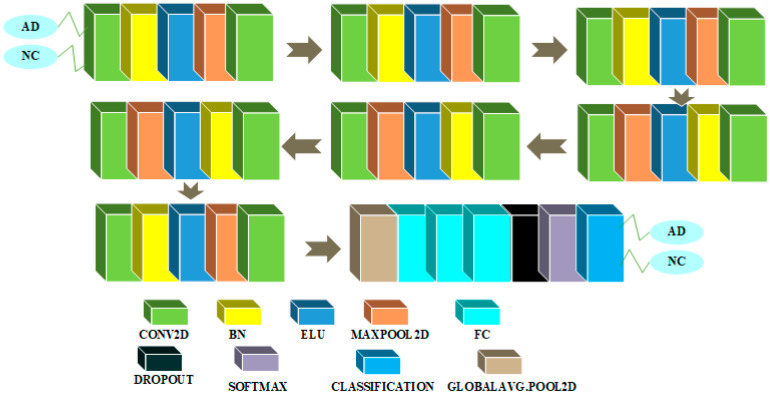
AD/NC binary classification architecture in the 2D domain.

**Figure 7 sensors-22-04609-f007:**
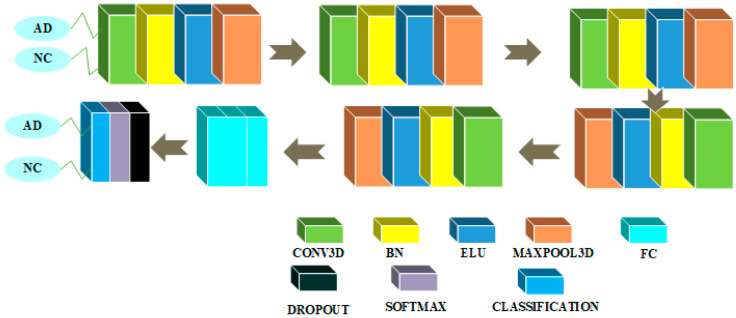
AD/NC binary classification architecture in the 3D domain.

**Figure 8 sensors-22-04609-f008:**
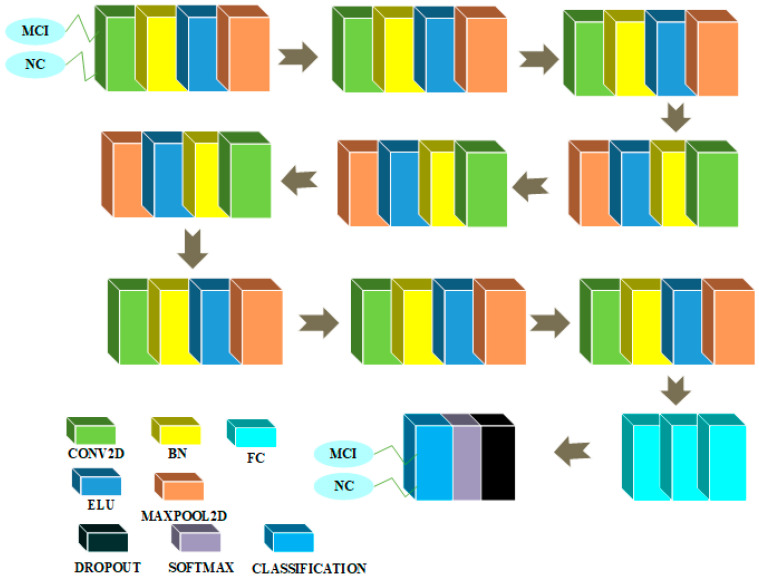
MCI/NC binary classification architecture in 2D domain.

**Figure 9 sensors-22-04609-f009:**
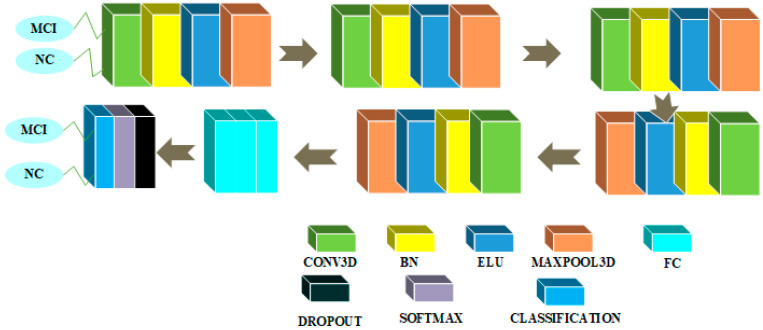
MCI/NC binary classification architecture in 3D domain.

**Table 1 sensors-22-04609-t001:** Demographics of patients and normal control individuals considered for this study.

Research Group	Number of Subjects	Age	Weight	FAQ Total Score	NPI-Q Total Score
NC	102	76.01 (62.2–86.6)	75.7 (49–130.3)	0.186 (0–6)	0.402 (0–5)
MCI	97	74.54 (55.3–87.2)	77.13 (45.1–120.2)	3.16 (0–15)	1.97 (0–17)
AD	94	75.82 (55.3–88)	74.12 (42.6–127.5)	13.67 (0–27)	4.07 (0–15)

**Table 2 sensors-22-04609-t002:** Performance metrics for the multiclass classification task.

Domain	Performance Metrics
3D	RCI = 0.2054,
	CEN = ‘AD’: 0.5088, ‘MCI’: 0.8038, ‘NC’: 0.5346,
	IBA = ‘AD’: 0.5660, ‘MCI’: 0.1091, ‘NC’: 0.5745,
	GM = ‘AD’: 0.7928, ‘MCI’: 0.4914, ‘NC’: 0.7406,
	MCC = ‘AD’: 0.5784, ‘MCI’: 0.1462, ‘NC’: 0.4614
2D	RCI = 0.03,
	CEN = ’AD’: 0.74, ’MCI’: 0.77, ’NC’: 0.76,
	IBA = ’AD’: 0.203, ’MCI’: 0.28, ’NC’: 0.1,
	GM = ’AD’: 0.574, ’MCI’: 0.51, ’NC’: 0.48,
	MCC = ’AD’: 0.22, ’MCI’: 0.029, ’NC’: 0.125

**Table 3 sensors-22-04609-t003:** Performance metrics for the AD-MCI binary classification task.

Domain	Performance Metrics
3D	SEN = 0.7021,
	SPEC = 0.7320,
	F_1_-score = 0.7097,
	Precision = 0.7174,
	Balanced Accuracy = 0.7170
2D	SEN = 0.5395,
	SPEC = 0.5976,
	F_1_-score = 0.5520,
	Precision = 0.5651,
	Balanced Accuracy = 0.5686

**Table 4 sensors-22-04609-t004:** Performance metrics for the AD-NC binary classification task.

Domain	Performance Metrics
3D	SEN = 0.8723,
	SPEC = 0.9118,
	F_1_-score = 0.8865,
	Precision = 0.9011,
	Balanced Accuracy = 0.8921
2D	SEN = 0.4288,
	SPEC = 0.6782,
	F_1_-score = 0.4823,
	Precision = 0.5511,
	Balanced Accuracy = 0.5535

**Table 5 sensors-22-04609-t005:** Performance metrics for the MCI-NC binary classification task.

Domain	Performance Metrics
3D	SEN = 0.5979, SPEC = 0.6471,
	F_1_-score = 0.6073, Precision = 0.6170,
	Balanced Accuracy = 0.6225
2D	SEN = 0.4729, SPEC = 0.5358,
	F_1_-score = 0.4823, Precision = 0.4921,
	Balanced Accuracy = 0.5043

**Table 6 sensors-22-04609-t006:** A comparative overview of methods on different binary and multiclass classification tasks for early AD diagnosis.

Authors	Data	Method(s)	Accuracy	Classification Task
Oh et al. [60]	MRI	Inception auto-encoder based CNN architecture	84.5%	AD/NC
Ekin Yagis et al. [61]	MRI	3D-CNN architectures	73.4%	AD/NC
Cosimo Ieracitano et al. [62]	MRI	Electroencephalo graphic signals	85.78%	AD/NC
**Proposed approach**	**PET**	**3D-CNN** **whole brain**	**89.21%**	**AD/NC**
Karim Aderghal et al. [63]	MRI	2D CNNs hippocampal region	66.5%	AD/MCI
Karim Aderghal et al. [64]	MRI	2D CNNs coronal, sagittal and axial projections	63.28%	AD/MCI
Firouzeh Razavi et al. [65]	MRI + PET + CSF	Scattered filtering and softmax regression	71.2%	AD/MCI
**Proposed approach**	**PET**	**3D-CNN** **whole brain**	**71.70%**	**AD/MCI**
Olfa Ben Ahmed et al. [66]	MRI	Circular Harmonic Functions	69.45%	NC/MCI
**Proposed approach**	**PET**	**3D-CNN** **whole brain**	**62.25%**	**NC/MCI**
Bijen Khagi et al. [67]	PET, MRI	DL architecture employing 3D-CNN layers	50.21%	AD/NC/MCI Multiclass
**Proposed approach**	**PET**	**3D-CNN** **whole brain**	**59.73%**	**AD/NC/MCI Multiclass**

## Data Availability

Not applicable.

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
