# Peer review of "Early-Stage Alzheimer’s Disease Categorization Using PET Neuroimaging Modality and Convolutional Neural Networks in the 2D and 3D Domains"

_sensors, 2022, doi:10.3390/s22124609_

Round 1

Reviewer 1 Report

In the present manuscript, the authors use convolutional neural networks as the roots of a supervised classification proposal for Alzheimer’s Disease (AD). The data used by the authors come from PET neuroimaging. There are numerous scenarios for comparison that are addressed by the authors including a 2D vs. 3D analysis, and a multiclass vs. binary class (pairwise) approach. As a result, and as one of the conclusions, the authors discuss that the 3D classification performs significantly better than the 2D. This performance seems to be better than current approaches in the literature. The authors also mention that blurring before subsampling is not helpful for 2D data as it is for 3D data.  

I think that the work is important, but I see several issues in the presentation of the results and in the discussion that should be addressed by the authors. The article fails in clarity on many points, and the authors should make an effort to improve around that. Please read below:

-       Abstract: Please avoid the word “basically” and make definitions more accurate.

-       Introduction: There are numerous phrases that the authors should rewrite/improve/correct. Some examples (but not all) are:

o   “Supervised classification is an important job in machine learning. “ -> Be more specific.

o   “Here aim is to” -> Here, the aim is to

o   “…and class imbalance etc.” -> remove “etc”

o   Another challenge is to present a solution that is compatible with real world settings. “ -> What do you mean by this?

o   “5G and beyond technologies” -> What is meant by “beyond technologies”?

o   The paragraph that starts with “Internet of Things” should include at least one citation.

o   The paragraph that is in red, that starts with “In the literature” is very confusing. There is not a single period in the entire paragraph and it looks more like a table than a paragraph. This should be rephrased with a better flow and not as a list of many different techniques.

-       Methods: where were the codes implemented?

-      Dataset Description: the authors should include links or citations to where the data can be found.

-       Methods: a discussion regarding how the data is divided into training and testing is missing. Also, are the authors using a validation dataset at any point?

-       Experiments: Why is it that two different approaches are used to compare 2D vs. 3D? In 2D you use stochastic gradient descent but in 3D you use Adam. If this is true, the comparison is not fair. Please revise.

-       Results and Discussion: All the measures indicated in the first paragraph here should be defined in the Methods. Either the authors should include the formulas or citations to where they can be found. In addition, a mention of which dataset they are taken from (training or testing) should be included.

-       Even though most of the results point to high accuracy, it seems to me that the one between MCI and NC has very poor accuracy, even in the case of the 3D domain. Is there a reason for that? Is there a way to overcome that?

Reviewer 2 Report

The paper needs improvements before it is published in a journal.

1. The contribution of the proposed algorithm should be presented in the introduction section.

2. The motivation behind the new methods is missing. The literature review and introduction sections are combined. Is there any reason? At the end of the literature, the need/motivation should be presented.

3. Section 2 is not clear whether existing datasets are used, or new datasets are generated. The existing sources need to be cited.

4. Parameters presented in section 5 on pages 9 and 10, need to be presented mathematically i.e., include equation.

5. Comparison is done with algorithms published in 2013, 15, and 16. Why recently published relevant algorithms are not included in the comparison?

6. The presentation of the results is poor i.e., the discussion on table 2 is followed by the discussion on other tables. After table 6 is discussed, table 2 is discussed again. Later, Table 6 is discussed again.

7. According to the results presented in Table 6, the proposed method achieves better accuracy in a few cases. What is the achievement of the proposed algorithm in this case? Clear justification is necessary that why in some cases the proposed algorithm works well and not in others.

8. I suggest naming the proposed algorithm or using the proposed instead of our method in Table 6.

9. The conclusion and the results and discussion section need to be consistent with each other.

1.        The paper needs thorough proofreading for typos and grammatical mistakes.

Round 2

Reviewer 1 Report

I thank the authors for answering all my questions and I think that the paper has improved. 

Reviewer 2 Report

Most of my concerns are addressed in the revised version. However, the results need to be discussed in more details with emphasize on the findings of the experiments. (if model A is performing better than model B the reasons should be discussed).
